

**Methane accumulation affected by particulate organic carbon in upper Yangtze**
**deep valley dammed cascade reservoirs, China**
Yuanyuan Zhang [1, 2], Youheng Su [2, 3], Zhe Li [1, 2*], Shuhui Guo[4], Lunhui Lu [1, 2],
Bin Zhang [2] , Yu Qin [3]
[1] College of Resources and Environment, Chongqing School, University of Chinese Academy of
Sciences, 400714, Chongqing, China
[2] CAS Key Lab of Reservoir Environment, Chongqing Institute of Green and Intelligent Technology,
Chinese Academy of Sciences, 400714, Chongqing, China
[3] College of River and Ocean Engineering, Chongqing Jiaotong University, 400074, Chongqing,
China
[4] Foreign Environmental Cooperation Center, Ministry of Ecology and Environment of the People's
Republic of China, 100035, Beijing, China

Corresponding author:
Email: lizhe@cigit.ac.cn



**Abstract**


Methane ($CH_4$) emissions from freshwaters to the atmosphere have a profound
impact on global atmospheric greenhouse gas (GHG) concentrations. Anthropogenic
footprints such as dam construction and reservoir operation significantly changed the
fate and transport of $CH_4$ in freshwaters. The type of particulate organic carbon (POC)
in reservoirs is a critical factor controlling $CH_4$ production and emissions. However,
little is known of how reservoir operation mediates the distribution of POC and
regulates $CH_4$ accumulation in cascade hydroelectric reservoirs. Here, spatial and
temporal variations in POC and $CH_4$ were explored in the Xiluodu (XLD) and
Xiangjiaba (XJB) reservoirs which are deep valley dammed cascade reservoirs located
in the main channel of the upper Yangtze River. Based on the $\delta^{13}C$-POC and N/C mole
ratios of particulate organic matter, the results of multi-endmember stable isotope
mixing models by a Bayesian model show that terrestrial POC and autochthonous POC
accounted for approximately $56 \pm 19\%$ and $42 \pm 19\%$ (SD, n=181) of POC, respectively.
$CH_4$ concentrations and $\delta^{13}C$-$CH_4$ in the cascade reservoirs were potentially influenced
by $CH_4$ oxidation. Together with other physicochemical parameters and structural
equation model, these results suggested that the input of terrestrial POC was dominantly
influenced by water level variations and flow regulation due to reservoir operation. The
cumulative effect of POC caused by cascade reservoirs was not apparent at a bimonthly
scale. Terrestrial POC was more likely to dominate $CH_4$ accumulation in cascade
reservoirs under reservoir operation.



## 1   Introduction


Methane ($CH_4$) is widely recognized as the second most important greenhouse gas
after carbon dioxide ($CO_2$) (Saunois et al., 2020). The latest data shows that global
atmospheric $CH_4$ was 1888.5 ppb in July 2021 (Dlugokencky, 2021). The annual
increase in global atmospheric $CH_4$ between 2007 and 2020 fell within the range of 7.99
$ppb·yr^{-1}$ to 14.81 $ppb·yr^{-1}$ (Dlugokencky, 2021). Global $CH_4$ emissions were estimated
to be up to 579 Tg. $yr^{-1}$ from 2008 to 2017 (Saunois et al., 2020). There is a very high
level of confidence that the atmospheric $CH_4$ increase during the Industrial Era was
caused by anthropogenic activities, which caused approximately 60% $CH_4$ emissions
(Ciais et al., 2014; Saunois et al., 2016; Saunois et al., 2020). However, not all the
sources of global $CH_4$ emissions are explicitly and well explained. Although half of the
global methane emissions come from the aquatic ecosystems (Rosentreter et al., 2021),
a large proportion of the uncertainties in the global $CH_4$ budget arise from freshwater
systems. The production of $CH_4$ in lakes and reservoirs is an important process in the
global methane cycle. This is partly because freshwater systems are closely linked to
and manipulated by anthropogenic activities, e.g., hydrological process regulation,
geomorphological alternation, large inputs of organic carbon, and nutrients from
surrounding communities. Anthropogenic footprints significantly change the fate and
transport of $CH_4$ in freshwaters.
Dam construction and reservoir impoundments are widely accepted as important
anthropogenic activities that significantly change the sink and sources of $CH_4$ in the
freshwater systems. In general, the net change in $CH_4$ emissions in reservoirs is
primarily contributed by the decomposition of organic matter (OM), e.g., soil organic
carbon and vegetation cover, due to flooding. Reservoir $CH_4$ emissions may reach the
highest levels immediately after impoundment and exponentially decline with aging
(Abril et al., 2005). Second, the reduced flow velocity and increased hydraulic retention
time (HRT) in the reservoir accumulates terrestrial OM from the upstream watershed to
the reservoir bottom, supporting the development of anoxic habitats at the reservoir

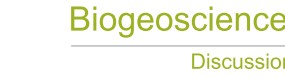

bottom for intensive methanogenesis. Site-specific hydro-morphological characteristics,
reservoir thermal regimes, and external OM loads from the upstream watershed
supported consistent reservoir $CH_4$ emissions. This evidence supported a wide
acceptance that reservoirs consistently emit $CH_4$ that exceeds those of upstream reaches
of the same river and, on average, from natural lakes (Stanley et al., 2016).

However, the role of dams and reservoirs in the global $CH_4$ budget has been

received challenges in recent years. Previous studies reported that global reservoirs
were either $CH_4$ neutral or $CH_4$ sources and that reservoir productivity and temperature
are better predictors than reservoir age (Deemer et al., 2016). Reservoirs may reduce
$CH_4$ and $CO_2$ emissions in downstream floodplain wetlands caused by upstream organic
carbon (OC) transport and sedimentation (Muller, 2019). Sediment loads and nutrient
enrichments were the primary and secondary driving factors that regulate $CH_4$
formation in navigable river impoundments (Wilkinson et al., 2019). While river
impoundments did not account for all hotspots of $CH_4$ formation in their study, they
implied that sediment management would offset $CH_4$ emissions from damming rivers
(Wilkinson et al., 2019). The largest share of $CH_4$ emissions may be due to OC retention
and mineralization in the reservoir (Shi et al., 2017). Recent research showed that
imbalanced stoichiometric sedimentation mediated $CH_4$ accumulation in the mid-part of
China's Three Gorges Reservoir (Li et al., 2020a). However, the findings supported that
reservoir operation strategy significantly impacts the patterns of $CH_4$ accumulation.
Growing research has highlighted the importance of reservoir hydrology and sediment
(or OC) dynamics contributing to $CH_4$ accumulation and emissions. If reservoir
operation could be proven to have an evident link with $CH_4$ dynamics and emissions,
there would be best practices for the hydropower industry to reduce excessive $CH_4$
emissions, which is meaningful for mitigating the global warming potential of
hydropower. However, the question of how reservoir operation mediates particulate
organic carbon (POC) transport and finally impacts $CH_4$ dynamics and emissions from
reservoir appears to be sophisticated and site-specific and has not been well addressed.

The study was extended from a single reservoir to a cascade system in the upper





Yangtze River to explore dissolved $CH_4$ accumulation under reservoir operation. The
research objectives are (1) to determine the input and accumulation of POC that
influence dissolved $CH_4$ accumulation under the physiochemical parameters caused by
cascade reservoir operation and (2) to explain how the different sources of POC regulate
the dissolved $CH_4$ accumulation in deep river-valley dammed reservoirs.

## 2 Methods

### 2.1 Site description and sampling campaign

The Xiluodu (XLD) reservoir and Xiangjiaba (XJB) reservoir are two deep
river-valley dammed cascade reservoirs located along the main channel of the upper
Yangtze River, which is frequently referred to as the "Jinsha River" (Figure 1). Both the
XLD and XJB hydro-projects serve as hydroelectricity production facilities. They also
perform significant seasonal water level adjustments for flood control as their partial
services (Figure 2). The initial impoundment of both reservoirs started in July 2013.
XJB finished its impoundment in September 2013, while XLD finished its full
impoundment one year later. The parameters of the XLD and XJB reservoirs are shown
in Table S1 (Li et al., 2017b).
The sampling campaign was conducted every other month between January 2018
and January 2019. Fifteen sampling sites were located along the main channel of both
reservoirs, L3 to L1 in the XLD reservoir, B10 to B1 in the XJB reservoir and X2 to X1
downstream of the XJB reservoir (Figure 1). Because it was not possible to cover all the
sampling sites and finish sampling work in one day, each sampling event was limited
between the dates of the 10th and 15th of that month. The sampling time of a day was
controlled between 8:00 AM and 6:00 PM. In particular, the inconvenience of local
transportation in such deep valley areas along the XLD reservoir limited sampling to 40
km upstream of the XLD dam, encompassing sampling sites L3 to L1. Downstream of
the XJB, to avoid tributary disturbance to collected samples, the sampling work was
limited within 10 km downstream of the XJB dam, encompassing sampling sites X2 and
X1.
10 L water samples were collected at 0.5 m below the water surface and



approximately 2 m above the sediment layer at each sampling site. However, in the river
reach right below both dams, where sites B10 to B8 and X2 to X1 were located, the
water column was well mixed and only surface water samples were collected. These
samples were then treated as representative of both surface and bottom samples in such
a fully mixed water column. The water temperature, dissolved oxygen (DO) and pH
were measured *in situ* by a calibrated YSI® Pro 2030 probe (YSI Inc., Ohio, USA).
Due to the different operation schemes of each reservoir, HRT was the
fundamental key variable structuring the aquatic ecosystem of both reservoirs and was
distinctive between the XLD and XJB reservoirs. A detailed calculation of HRT is
provided in the Supporting Information (Section S1.1). Considering 2018 as an example
(Figure S1), the annual average cumulative HRT in the XLD reservoir was 32.9 ± 20.6
days (mean ± SD), with the 1st and 3rd quartiles between 14.7 days and 46.4 days. In
the XJB reservoir, the corresponding data collected in 2018 was 15.8 ± 8.6 days, with
1st and 3rd quartiles between 6.9 days and 23.4 days. The spatial and temporal
variations in the cumulative HRT in the XLD and XJB reservoirs also showed that the
cumulative HRT was positively correlated with the water level and negatively
correlated with the flow and shortest in the flood season (Figure S2).



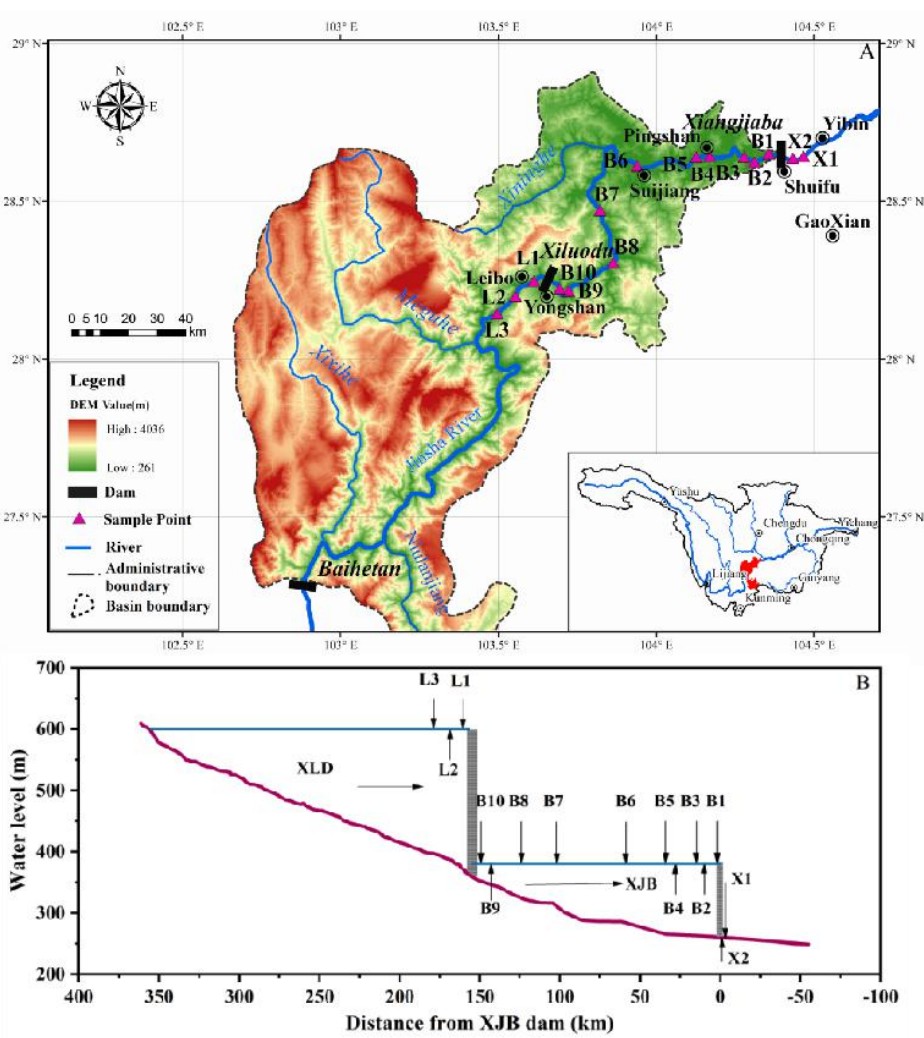

**Figure 1. The Xiluodu (XLD) and Xiangjiaba (XJB) cascade reservoirs. A) the map of the two reservoirs and sketch of sampling sites; B) vertical profiles of the cascade reservoirs and location of the sampling sites.**

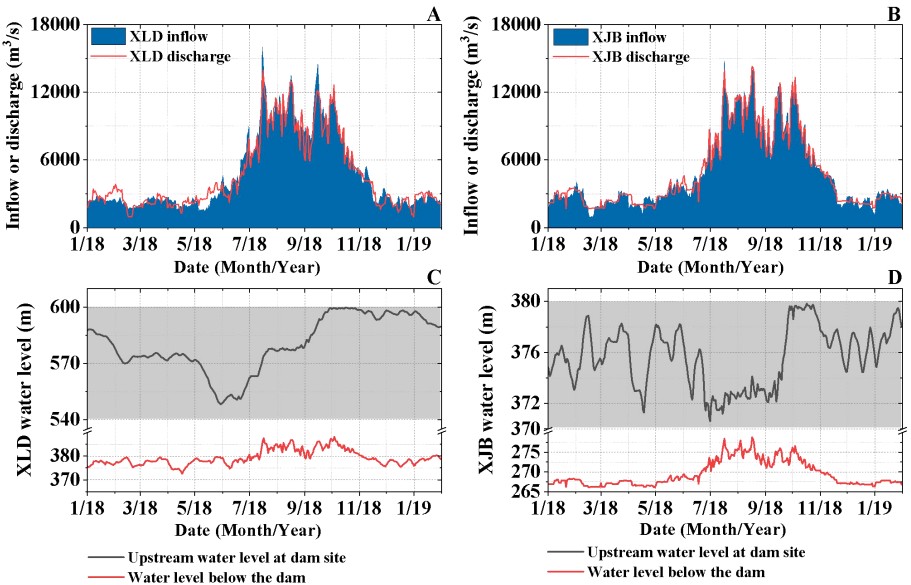

**Figure 2. A) and B) daily reservoir inflow and discharge of the Xiluodu (XLD) and Xiangjiaba (XJB) reservoirs during the study; C) and D) daily water level variations at the upstream and downstream of the two dams. Daily reservoir inflow/discharge and water level variations of each reservoir were provided by the China Three Gorges Corporation (www.ctg.com.cn).**

## 2.2 Measurement of CH$_4$ and CO$_2$ concentrations and other environmental parameters

CH$_4$ and CO$_2$ concentrations in the water phase were analyzed by the headspace approach (Unesco/Iha, 2010). Equipped with a Picarro® G2201-$i$ Isotopic Analyzer (California, USA), our standard operation procedure was upgraded to the CH$_4$ and CO$_2$ headspace approach in 2017. Water samples were carefully collected by a 300 mL polypropylene syringe barrel and sealed underwater by its matching piston. Then, surplus water and possible air were ejected out carefully until the water sampling was 200 mL in the syringe. Then, 100 mL of highly purified nitrogen was sucked into the syringe and shaken immediately on-site for 2 minutes for gas exchange. Then, 100 mL gas samples were injected into pre-purified air sampling bags (0.3 L, HEDE tech, Dalian, China) for further analysis by Picarro® G2201-$i$ Isotopic Analyzer under HP mode. Triplicates were performed for quality control.

For dissolved nutrients, water samples were filtered through pretreated (combusted



at 450 °C for 4 h in a muffle furnace and weighed after cooling) Whatman® GF/F glass
fiber membranes (Whatman®, UK). Dissolved organic carbon (DOC) was then analyzed
by a Shimadzu® TOC-V TOC analyzer (Shimadzu, Japan). Dissolved total nitrogen
(DTN) and total phosphorus (DTP) were analyzed by the standard method (Wef and
Apha, 2005).
For particulate matter, the chlorophyll a (Chl-a) concentration was measured
spectrophotometrically after 90% cool acetone extraction. All residues on the GF/F
glass fiber membranes were dried at 65 °C for 48 h and reweighed. The concentration of
total particulate matter (TPM) in the water column was the quotient of the mass
difference between the two weights and the volume of water samples used infiltration.
The 65 °C dried fresh residues were also used for elemental composition analysis of C,
and N. POC, particulate organic nitrogen (PON), $\delta^{13}$C-POC and $\delta^{15}$N-PON were
measured by a Thermo Fisher ® Flash H T Elemental Analyzer for Isotope Ratio MS
(Thermo Fisher Scientific, MA, USA). The standard reference materials were Vienna
Pee Dee Belemnite for carbon and atmospheric $N_2$ for nitrogen. Triplicates were also
performed for quality control.
**2.3 Application of stable isotope mixing models**
Fits Stable Isotope Mixing Models (SIMMs) were meant as a longer-term
replacement to the previously widely used package SIAR (Stable Isotope Analysis in R)
(Parnell et al., 2010). SIMMs were used to infer dietary proportions of organisms
consuming various food sources from observations on the stable isotope values taken
from the organisms' tissue samples. The proportional contribution of different sources
of OM to a mixture was estimated by using stable isotope mixing models " SIMMs ",
which are based on Bayesian methods (Parnell et al., 2010; Parnell et al., 2013).
The newest Simmr package of stable isotope mixing models was implemented in
the R program (Version: 4.0.3) to estimate the contribution of different sources of POC.
The contributions of different endmembers to POC were calculated by the $\delta^{13}$C, $\delta^{15}$N
and N/C molar ratios of particulate organic matter (POM). To estimate the contribution
of POC sources, the endmember values of C3 and C4 plants were set from references
(Kendall et al., 2001; Wu et al., 2007; Jiang and Ji, 2013; Wang et al., 2014; Chen et al.,
2018b; Deng et al., 2018; Xuan et al., 2019; Ru et al., 2020), and the endmember values
of soil and plankton from the upper Yangtze River were sampled and analyzed in the
field sampling campaign (Table 1). The 50% quantile values were the predicted values
of the model. The contribution of terrestrial POC was obtained by C3 and C4 plants,
coastal soil, and plankton as the contribution of autochthonous POC.
**Table 1. $\delta^{13}$C, $\delta^{15}$N, N/C molar ratios of particulate organic matter (POM) from different**
**endmembers**

| Endmember | $\delta^{13}$C-POM (‰) | $\delta^{15}$N-POM (‰) | N/C |
|-----------|-----------------------|-----------------------|-----|
| C3 plant | -28.9 ± 1.6 | 1.8 ± 3.7 | 0.038 ± 0.019 |
| C4 plant | -13.4 ± 1.0 | 2.4 ± 5.7 | 0.038 ± 0.019 |
| Soil | -21.0 ± 2.0 | 4.6 ± 1.4 | 0.134 ± 0.022 |
| Plankton | -26.8 ± 0.6 | 6.6 ± 1.8 | 0.184 ± 0.011 |

**2.4 Data analysis**
The structural equation model was performed using the Lavaan package in the R
program to obtain the effects of different variables by verifying the theoretical model.
Since the original data difficultly satisfied the normal distribution tested by
Kolmogorov–Smirnov test, the maximum likelihood relaxation algorithm (MLR) was
used as the estimator.
Data analyses and plotting (Kolmogorov–Smirnov test, linear regression, and one-
way analysis of variance [ANOVA]) were performed using OriginPro® 2018
(OriginLab Corporation, MA, USA) and Statistical Product and Service Solutions
(SPSS). In one-way ANOVA, differences between means and 50% quantile values were
considered significant at $p < 0.05$. A detailed description of the mass balance approach
of the POC and $CH_4$ in both reservoirs is provided in the Supporting Information
(Section S1.1).



## 3  Results

### 3.1 Physical limnology and major environmental parameters

The water temperature ranged from 12.45 °C to 25.9 °C and 13.6 °C to 26.3 °C in the XLD and XJB reservoirs, respectively. The thermal stratification patterns of both reservoirs are different (Figure S3A). Thermal stratification developed in the XLD reservoir in spring, e.g., March, and persisted throughout the whole summer. Metalimnion existed approximately 60 m below the water surface. With the increase in water level due to reservoir impoundment, together with the decrease in air temperature, thermal stratification gradually disappeared in November. In the XJB reservoir, weak thermal stratification was initiated in March, which supported its spring algal blooms (Figure S3C). However, because the upstream reservoir increased its discharge to prepare abundant reservoir capacity before the summer flood season, the XJB reservoir was fully mixed in May. Hypolimnion in the XJB reservoir was limited to approximately 150 m below the water surface in May and approached to 180 m below the water surface in July as the discharge of the XLD reservoir apparently increased. In September, the XJB reservoir was fully mixed.

The DO in all samples varied from 7.06 to 16.08 mg/L with a mean value of $9.23 \pm 0.91$ mg/L (SD, n=181). The DO of surface water was significantly higher than that of bottom water in the XJB reservoir due to algal blooms in March. Additionally, the DO of the XLD reservoir in the flood season was slightly lower than that in the dry season (Figure S3B). Although both reservoirs were impounded in less than a decade, the trophic status of both reservoirs is meso-oligotrophic. The chlorophyll a (Chl-a) concentrations varied widely, with values of 0.01–35.02 mg/m$^3$. The maximum Chl-a concentration in surface water during the study was found in the XJB reservoir. The diatom bloom initiated in late February and continued to April in the XJB reservoir. The Chl-a concentration in the XLD reservoir was below 15 mg/m$^3$ throughout the year (Figure S3C). The bloom-forming period of the XLD reservoir was approximately the same as that of the XJB reservoir.



The DOC concentrations in all samples varied from 0.04 to 6.13 mg/L with an
average value of 2.37 ± 1.24 mg/L (SD, n=175). The maximum DOC concentration was
found between May and July, with a mean value of 3.94 ± 0.69 mg/L (SD, n=52), and
the minimum average value was found in January (1.24 ± 0.75 mg/L, SD, n=45, Figure
S3D). There was no obvious spatial difference in DOC concentrations between the two
reservoirs, which indicates that the DOC concentrations may be significantly affected
by the input of upstream. In all the sampling sites, the TN and TP in the water column
were 146.78 ± 64.6 μM (SD, n=182) and 2.51 ± 1.12 μM (SD, n=156), respectively.
The mole ratio of TN: TP, as an indicator of nutrient limitation, fell in a range between
42.93 (1st quartile) and 91.84 (3rd quartile), indicating heavily P limitation for primary
producers in both reservoirs.
**3.2 Particulate organic matter and stable isotopic signatures**
The TPM concentrations in all sampling sites of both reservoirs varied from 1.10
to 38.80 mg/L with a mean of 5.74 ± 5.03 mg/L (SD, n=182), and the average maximum
TPM concentration was found in July and September with a value of 10.78± 4.34 mg/L
(SD, n=52). The POC concentrations in all water samples varied from 0.03 to 2.81 mg/L
with a mean of 0.35 ± 0.39 mg/L (SD, n=182, Figure 3). The PON concentrations in all
samples ranged from 0.01 to 0.21 mg/L, with a mean of 0.04 ± 0.03 mg/L, and the N/C
mole ratios varied from 0.019 to 0.273, with an average value of 0.116 ± 0.032 (SD,
n=182, Figure 3). The maximum POC and PON concentrations and minimum N/C mole
ratios were found in March, with average values of 0.98 ± 0.64 mg/L, 0.07 ± 0.05 mg/L
and 0.079 ± 0.052, respectively (SD, n=26, Figure 3). Meanwhile, the maximum POC
and PON concentrations in the surface and bottom water were found before the XJB
dam, and the N/C mole ratios from upstream were lower than those downstream during
this period. In particular, the values in the surface water were different from those in the
bottom water, which may be due to the lower flow and temperature stratification (Figure
3A, 3B).



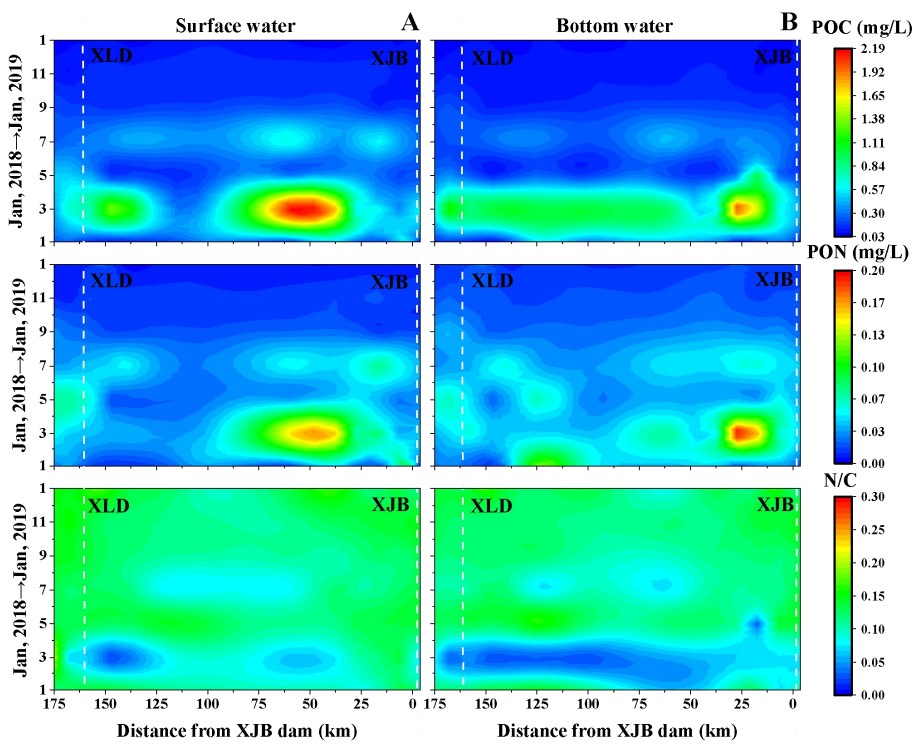

**Figure 3. Wavelet figure showing the spatial and temporal distributions in particulate organic carbon (POC), particulate organic nitrogen (PON), and N/C mole ratios of particulate organic matter of the surface (A) and bottom (B) water from the XLD and XJB reservoirs.**

The stable isotopic ratios and N/C mole ratios of POC and PON reveal that POM originated from different sources in the XLD and XJB reservoirs. Most $\delta^{13}$C-POC values in all samples showed no significant differences during the year ranging between −13.0‰ and −35.3‰, with a mean of −26.1 ± 1.8‰ (SD, n=181). However, $\delta^{13}$C-POC in March showed obvious differences that were higher than other values (−24.3 ± 3.3‰, SD, n=26). $\delta^{15}$N-PON varied from −1.5‰ and 17.5‰ with a mean of 6.5 ± 2.7‰ (SD, n=182) which displayed irregularly high values in January 2018 (8.9 ± 4‰, SD, n=26). This may be influenced by the inputs of excrement or other anthropogenic activities during this period.

As shown in Figure 4A to Figure 4C, most POM in the XLD and XJB reservoirs may have been derived from plankton. However, the $\delta^{15}$N-PON, which may be affected by trophic levels in the two reservoirs displayed large fluctuations (Chen et al., 2018a).





Thus, the N/C mole ratios of POM and $\delta^{13}$C-POC can better constrain the contribution
of different sources of POC in this study. As the average $\delta^{13}$C-POC ($-26.1$‰) and N/C
mole ratios of POM (0.116) were similar to those of plankton, with values of $-26.8$‰
and 0.184 respectively (Table 1), POC in the XLD and XJB reservoirs may have been
primarily from plankton ($42 \pm 19\%$) and secondarily, from C3 plants ($34 \pm 16\%$) and
less from soil ($13 \pm 11\%$) and C4 plants ($8 \pm 9\%$) (SD, n=181) (Figure 4D). Previous
studies have reported that particulate matter derived from erosion of soil may have low
OC adsorption due to the low clay content of surface soils in the Jinsha River Basin
(Wu et al., 2020), which agrees with the results of this study. Conversely, the POC/Chl-a
threshold ratio of 300 indicates the dominance of phytoplankton in the POC pool
(Suzuki et al., 2014; Kang et al., 2019; De Castro Bueno et al., 2020), and more than
half of the samples in the surface water were classified with the origin of phytoplankton
dominance (Figure S4), which is similar to the results of stable isotope mixing models.
POC can be divided into autochthonous and terrestrial POC. Autochthonous POC
mostly originates from plankton, yet terrestrial POC derives from plants and soil in the
catchment (Guillemette et al., 2013; Chen et al., 2018b; Tittel et al., 2019). The
concentrations of autochthonous and terrestrial POC provided in the Supporting
Information (Section S1.2, Figure S5) were estimated by the contributions of plankton,
soil, C3 and C4 plants.



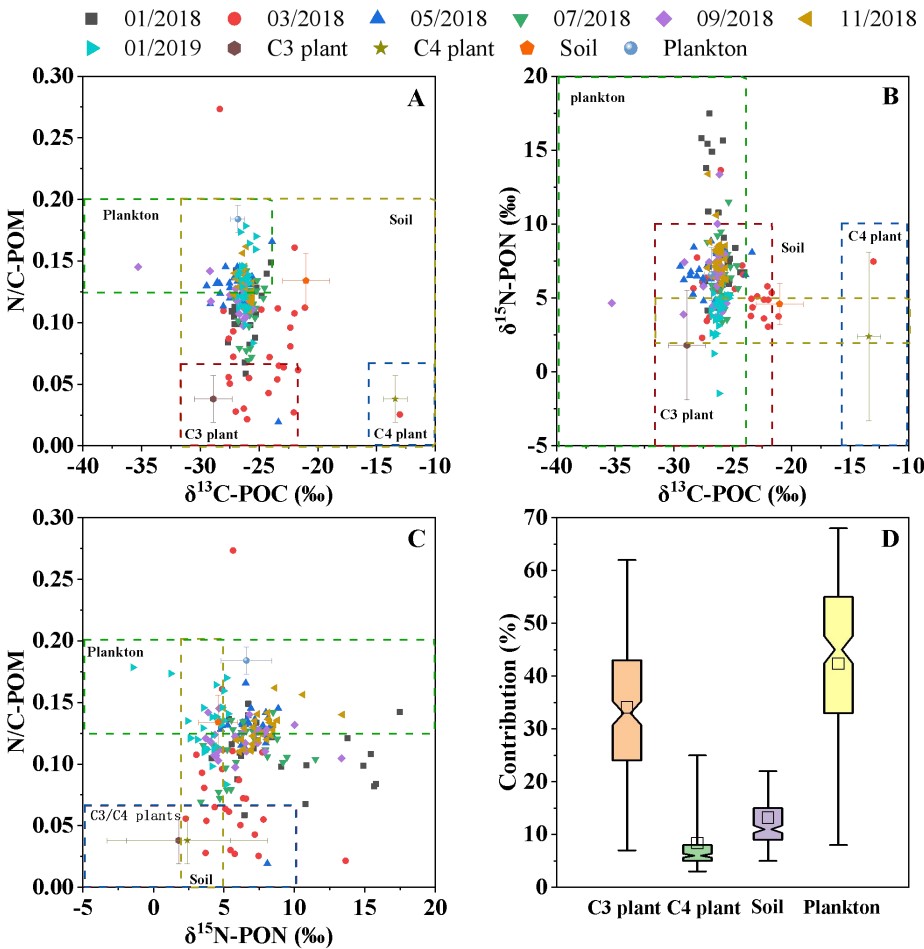

**Figure 4. A) N/C mole ratios vs. δ¹³C-POC, B) δ¹⁵N-PON vs. δ¹³C-POC, C) N/C mole ratios vs. δ¹⁵N-PON in particulate organic matter (POM) and endmembers (C3 and C4 plants, soil, plankton) (the dotted area in A, B and C was from literature (Kendall et al., 2001)); D) the contribution of endmembers in the XLD and XJB reservoirs.**

### 3.3 Spatial and temporal variations in $CH_4$ and $CO_2$ concentrations, $δ^{13}C$-$CH_4$

The annual $CH_4$ and $CO_2$ concentrations in all water samples from the two cascade reservoirs varied from 0.011 to 0.133 μmol/L and 0.016 to 0.257 mmol/L, with average values of 0.039 ± 0.024 μmol/L and 0.046 ± 0.025 μmol/L (SD, n=182), respectively. The annual average $CH_4$ concentration observed in the study was lower than those in global rivers and streams (1.45 ± 7.98 μmol/L, SD, n=1439) (Stanley et al., 2016). In XLD reservoir (L3–L1), annual $CH_4$ fell in a range between 0.014 μmol/L and





0.052 μmol/L, with a mean of 0.026 ± 0.008 μmol/L during the study (SD, n=42)
(Figure 5). Annual $CO_2$ in XLD reservoir fell in a range between 0.033 mmol/L and
0.092 mmol/L with a mean of 0.046 ± 0.017 mmol/L (SD, n=42) (Figure 5). In XJB
reservoir (B10–B1), annual $CH_4$ fell in a range between 0.012 μmol/L and 0.133
μmol/L with a mean of 0.044 ± 0.026 μmol/L (SD, n=126). Correspondingly, annual
$CO_2$ in the XJB reservoir fell in a range between 0.016 mmol/L and 0.257 mmol/L with
a mean of 0.047 ± 0.028 mmol/L. Downstream of the XJB reservoir (X1–X2) had a
range of $CH_4$ between 0.011 μmol/L and 0.064 μmol/L with a mean of 0.033 ± 0.016
μmol/L (SD, n=14). The water column $CO_2$ concentration downstream of the XJB
reservoir was between 0.030 mmol/L and 0.077 mmol/L with a mean of 0.041 ± 0.013
mmol/L (SD, n=14). The longitudinal gradient of $CH_4$ concentration from upstream
XLD reservoir to downstream XJB reservoir was evident, and the mid-part of the XJB
reservoir (B6–B4) showed a relatively high level of $CH_4$ concentration among all
samples. Comparatively, the longitudinal gradient of $CO_2$ concentration along the
cascade reservoir was not apparent (Figure 5). The mole ratios between $CH_4$ and $CO_2$
($CH_4/CO_2$)were among the highest in the middle of the XJB reservoir. For temporal
variations, the highest $CH_4$ concentration at the dam site of the XLD reservoir was
shown in the flood season. However, the peak values of $CH_4$ concentrations at both the
surface and bottom water of the XJB reservoir were observed during the flood season.
However, there was no significant difference in $CH_4$ concentrations between the surface
and bottom water. The lowest $CO_2$ concentration was found before the XJB dam in
March, and the $CO_2$ concentrations in surface water were significantly lower than those
in bottom water (Figure 5A, 5B).

Stable carbon isotopic signatures of $CH_4$ in the water column could help to

explore and identify $CH_4$ production and transformation in reservoirs (Whiticar and
Faber, 1986; Lima, 2005; Templeton et al., 2006). $\delta^{13}C$-$CH_4$ in all sampling water of
both reservoirs ranged from −8.3‰ to −54.0‰ with a mean of −32.7 ± 9.0‰ (SD,
n=156, Figure 5). $\delta^{13}C$-$CH_4$ in all water samples showed obvious temporal and spatial
variations, and maximum values were found before and under the two dams. Lower



δ13C-CH4 values were observed in March and July, where the peak value of CH4
concentrations was found. However, most δ13C-CH4 in both surface and bottom water
from the two reservoirs displayed no significant differences except for that in March
from the XJB reservoir (Figure 5A, 5B).

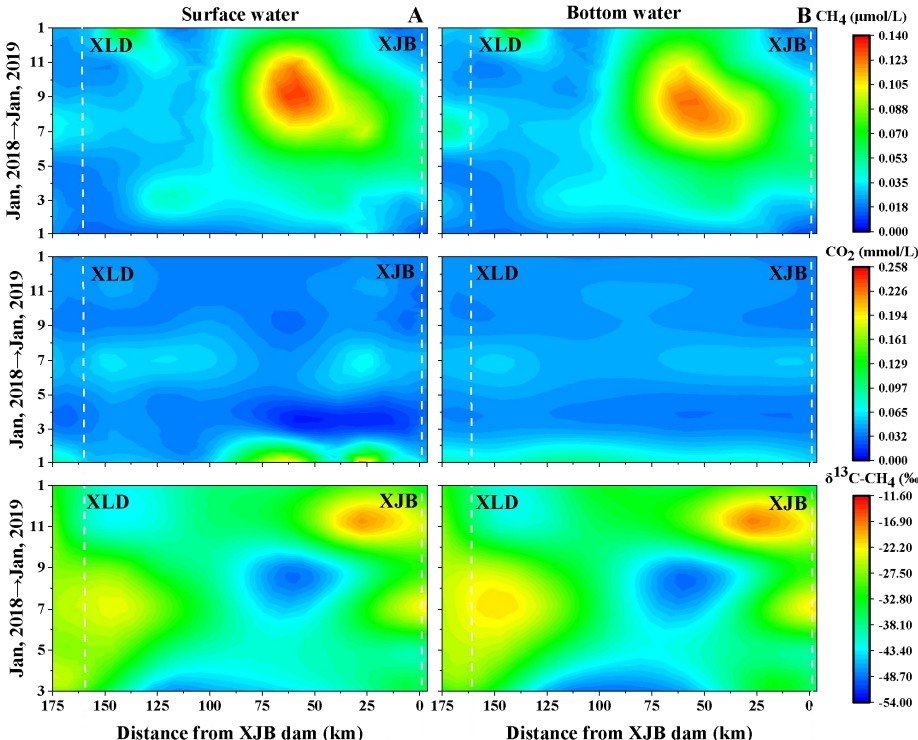


**Figure 5. Wavelet figure showing the spatial and temporal distributions in CH4, CO2 and**
**δ13C-CH4 of the surface (A) and bottom (B) water from the XLD and XJB reservoirs.**
In freshwaters, a large fraction of CH4 in the aquatic food network exists as a
carbon source which was oxidized by bacteria, can reduce CH4 entering the water by
diffusion (Jones and Grey, 2011; Taipale et al., 2011; Frossard et al., 2015; Sawakuchi et
al., 2016; Matoušů et al., 2019; Thottathil et al., 2019; Saarela et al., 2020), which may
change CH4 dynamics in water. Previous studies reported that the methane oxidizing
bacteria (MOB) can survive at oxic-anoxic interfaces in freshwater systems, and CH4
oxidation was frequently found in environments with high CH4 and DO concentrations
(Bagnoud et al., 2020; Reis et al., 2020; Bai et al., 2021). Moreover, the δ13C values of



biogenic methane during methanogenesis were very low, such as methanogenesis by
acetate fermentation in freshwaters, and the $\delta^{13}$C-CH$_4$ was then gradually elevated
during CH$_4$ oxidation due to $^{13}$C enrichment in residual CH$_4$ (Whiticar and Faber, 1986;
Conrad, 1999; Bastviken et al., 2002; Lima, 2005). The magnitude of $\delta^{13}$C-CH$_4$ by
MOB significantly depends on environmental conditions, so the fractionation factor ($\alpha$)
caused by microbial CH$_4$ oxidation shows a large range of 1.003 to 1.039 (Templeton et
al., 2006).

The study area was divided into three sections along longitudinal gradients,

B1-B4/L1-L3 as lacustrine regions (before the dam), B5-B7 as transition regions, and
X1-X2/B8-B10 as riverine reaches. In this study, there was no apparent anoxic water
column in the lacustrine regions (upstream of the dam) of the XJB reservoir. Therefore,
the oxic surface sediment and large water depth could have mitigated the release of CH$_4$
bubbles potentially produced by the deep sediment, which was likely to lead to low CH$_4$
accumulation in water before the dam. The most $\delta^{13}$C-CH$_4$ in the water column were
above −50‰ of methanogenesis by acetate fermentation, which indicated that CH$_4$ was
dominantly oxidized in the two reservoirs, especially $\delta^{13}$C-CH$_4$ in the lacustrine region,
had higher values and was significantly different from riverine reaches and transition
regions (Figure S6). Therefore, the CH$_4$ accumulation in the water column may be
influenced by MOB. Significantly, the most CH$_4$ oxidation process from B6 (the hotspot
of CH$_4$) to B1 (before dam) in the XJB reservoir can also be presented by Rayleigh
fractionation, for example, CH$_4$ concentrations and $\delta^{13}$C-CH$_4$ in the March, July,
September 2018 and January 2019 satisfied the fractionation caused by microbial CH$_4$
oxidation, and $\alpha$ ranged from 1.01 to 1.035 (Figure S7). The CH$_4$ oxidation process was
interestingly discovered along with the river flow, but the $\delta^{13}$C-CH$_4$ in vertical water
showed no significant differences. Only the $\delta^{13}$C-CH$_4$ in bottom water from the XJB
reservoir was significantly lower than that in surface water in March, yet there was no
significant difference in CH$_4$ concentrations (Figure 5). Although recent studies have
indicated that the $\delta^{13}$C values of CH$_4$ produced in oxic waters are usually less negative
than those of CH$_4$ accumulation in an anaerobic environment (Hartmann et al., 2020),



more evidence is needed to support this process in this study.

## 4    Discussion

### 4.1 Cascade damming effect on input and accumulation of POC

The distribution of POC in a river can be significantly influenced by dams. At first,
damming increased losses of river connectivity and reduced water flow in the main
channel, which possibly decreased the transport of POC after damming due to the long
HRT, further resulting in bulk accumulation and burial of POC in reservoirs (Ulseth and
Hall Jr, 2015; Li et al., 2017a; Almeida et al., 2019; Li et al., 2020a; Wang et al., 2020).
Additionally, dam construction promotes the intensive water-level fluctuations to cause
landslides and debris flows and indirectly influence land types, which may increase the
terrestrial POC in the reservoir (Yao et al., 2006; Luo et al., 2016; Iqbal et al., 2018;
Zorzal-Almeida et al., 2018).
Based on the N/C mole ratios and $\delta^{13}$C-POC values, POC mostly originated from
terrestrial POC in the XLD and XJB reservoirs. Water temperature was the only factor
that exhibits an obvious positive effect on autochthonous POC (Figure S8), which was
mainly caused by the outbreak of algal blooms in spring to increase autochthonous POC
in summer (Figure S5). The water level of the XJB reservoir and water flow exhibit a
significantly positive effect on terrestrial POC (Figure S8). Due to intensive water-level
fluctuations, the high-water level in January was modulated to a low water level in
March by reservoir scheduling operation, which led to an increase in the input of
terrestrial POC in March (Figure 2C, 2D, S5). Additionally, terrestrial POC in July
under high water flow displayed an increasing trend that was possibly affected by runoff
erosion. The water level and water flow caused by reservoir scheduling operation were
primary factors for the input of terrestrial POC. Meanwhile, due to the low clay content
of surface soil in the river basin, the particulate matter derived from soil erosion may
have a low OC content (Yu et al., 2011; Wu et al., 2020). This study showed that the
proportion of POC in TPM during the flood season was lower than that in the dry
season in cascade reservoirs (Figure S9). These results imply that particulate matter





from the river basin was less composed of OC, which suggests that a small amount of
terrestrial POC possibly enters the XLD and XJB reservoirs during the flood season.

The DOC/POC ratios with a global average value of 1-1.2 displayed an

exponentially negative correlation with the deposition of particulate matter in major
rivers (Ran et al., 2013). In this study, the DOC/POC ratios in March from the XLD and
XJB reservoirs with an average value of 3.96 were significantly lower than those in the
other months (Figure 6) and close to the Long Chuan River in the Jinsha River Basin
(Lu et al., 2012), which suggests that the sedimentation of POC in March was
significant. However, the average DOC/POC ratio in the flood period was higher than
10, which indicates that the accumulation of POC in the flood period was very low.
Because there are no large tributaries in the XJB reservoir, the accumulation of POC
from upstream and riverbanks were primarily reflected by the mass balance calculation
of POC (Figure 7A). According to the mass balance results of POC, the accumulation of
POC was mainly reflected in March (Figure 7A), while the difference in POC flux in
July and September during the flood period was not evident. These results also indicate
that the accumulation of POC in the XJB reservoir was limited during the flood season.
Therefore, the cumulative effect of POC at a bimonthly scale was not significant in the
XLD and XJB reservoirs.



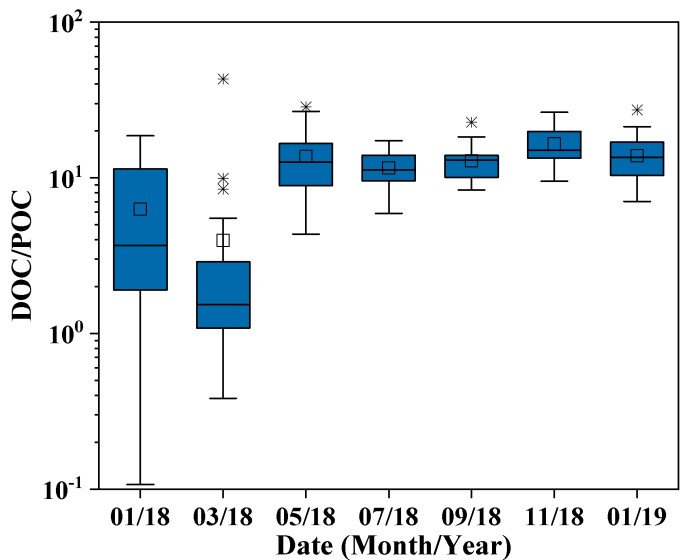


**Figure 6. Box plot showing the distribution of DOC/POC ratios in sampling month from the XLD and XJB reservoirs (the asterisks, black squares and middle black lines represented outliers, mean values and medians).**




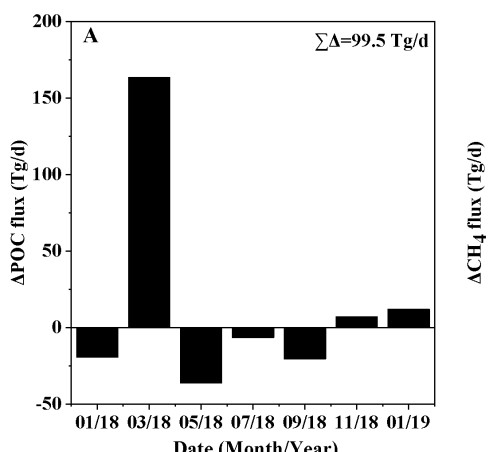
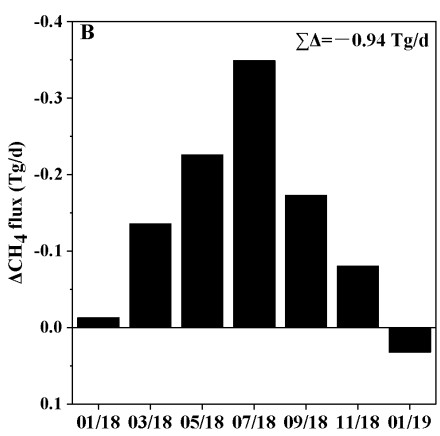


**Figure 7. Mass balance calculations of POC (A) and CH$_4$ (B) in sampling month from the**
**XJB reservoir.**
**4.2 CH$_4$ accumulation influenced by terrestrial POC in deep river-valley dammed**
**cascade reservoirs**
CH$_4$ production and emissions in a freshwater system can be affected by
parameters such as geography, hydrology, OM, water temperature, and final electron



receptors, yet POC was the most important factor for $CH_4$ accumulation in rivers
(Baulch et al., 2011; Crawford et al., 2014; Stanley et al., 2016). The degradation of
different sources of POC in freshwaters was different. Autochthonous POC has been
found to have a high bioavailability, yet terrestrial POC was already partly degraded and
less available for microorganisms (Guillemette et al., 2013). Autochthonous POC can be
the main reason for promoting $CH_4$ production in freshwaters, such as Lake Fuxian and
Lake Diamond in the USA (West et al., 2012; Li et al., 2020b). Due to terrestrial POC
accounting for large proportions in sediment, terrestrial POC can also have a potential
promotion effect on $CH_4$ accumulation, such as the William H. Harsha reservoir in the
USA (Berberich et al., 2020) and the Congo River (Upstill-Goddard et al., 2017).
Therefore, the river reservoirs affected by dam interception have complicated
hydrological characteristics of lakes and rivers, which results in different impacts of
autochthonous and terrestrial POC on $CH_4$ in reservoirs.

Young reservoirs are frequently believed to be significant $CH_4$ emitters due to the
decomposition of flooded OC (Prairie et al., 2018). The structural equation model can
be used to explore the relationship between different sources of POC and $CH_4$ by
various physicochemical factors (Figure S8). In terms of exogenous variables, the
hydropower production of the XJB reservoir, water temperature and flow show positive
correlations, and the water level of the XJB reservoir displayed negative correlations
with the hydropower production of the XJB reservoir, water flow, and temperature.
Additionally, $CH_4$ accumulation in deep river-valley dammed cascade reservoirs was
positively affected by terrestrial POC, water flow, and temperature rather than
autochthonous POC, hydropower production, and the water level of the XJB reservoir
(Figure S8). However, this path model cannot illuminate the relationship among $CH_4$
($R^2=0.22$), autochthonous ($R^2=0.17$) and terrestrial POC ($R^2=0.11$) in the two reservoirs.
This implies that the causality among variables in the temporal dimension is not
synchronous, and more factors affecting relationships among $CH_4$ concentration and
autochthonous and terrestrial POC in the spatial dimension need to be considered.

The dam constructions hindered the water velocity, resulting in increased HRT



(Wang et al., 2020). The relationships among HRT, $CH_4$ concentration, autochthonous
and terrestrial POC are discussed below. The terrestrial POC and $CH_4$ concentrations in
the XLD and XJB reservoirs present single peaks with HRT (Figure 8A, 8C), and
autochthonous POC displays a positive correlation with HRT (Figure 8B). HRT as an
exogenous variable in the structural equation model, was also used to explain the spatial
relationship among endogenous variables such as $CH_4$ and autochthonous and terrestrial
POC (Figure 9). The structural equation model shows a significant positive effect
between HRT and autochthonous POC, which can be explained by an increase in
phytoplankton photosynthesis (Crawford et al., 2016).





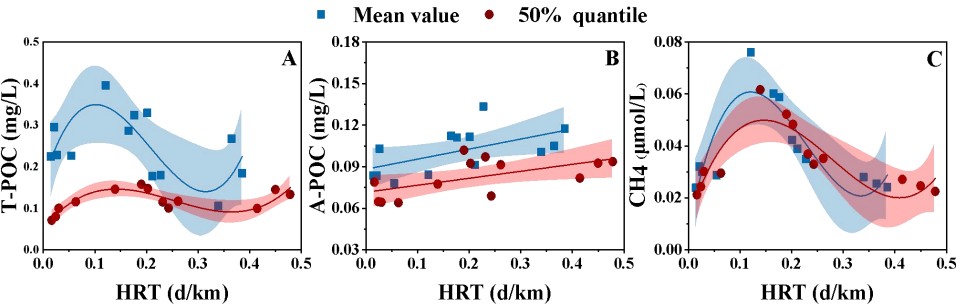

**Figure 8. Polynomial and linear fit relationship between POC, CH$_4$ and HRT in the spatial dimension from the XLD and XJB reservoirs (the blue and red area represented confidence bands and confidence level for curves). The mean values and 50% quantiles of these variables were considered.**

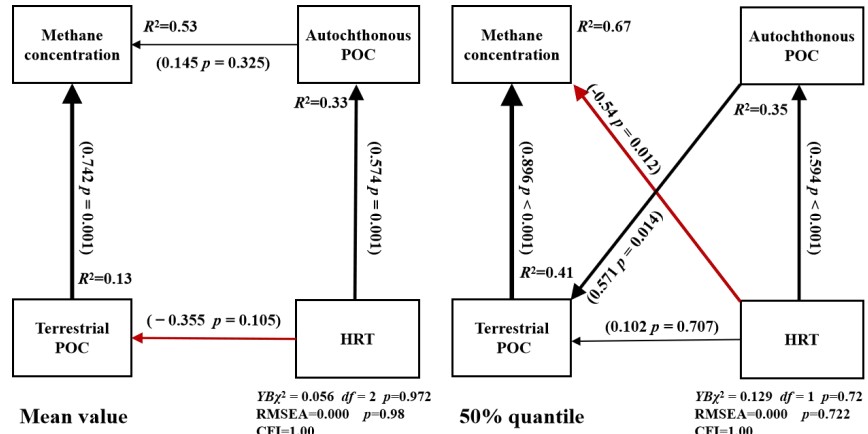

**Figure 9. Structural equation model among the mean values (left) and 50% quantiles (right) of HRT, autochthonous POC, terrestrial POC and CH$_4$ in the spatial dimension from the XLD and XJB reservoirs.**

Previous investigations have shown that autochthonous POC in lake reservoirs with depths greater than 100 m was completely mineralized before carbon burial (Steinsberger et al., 2020). Autochthonous POC can be rapidly used to produce CH$_4$, while the decomposition of terrestrial POC was slower (Grasset et al., 2018; Isidorova et al., 2019). Autochthonous POC can be easily oxidized in deep water columns before entering anaerobic environments or less distributed in surface sediments. Additionally, surface sediments often do not have good stability and an anaerobic environment, which results in autochthonous POC being more likely to be converted to CO$_2$ (Isidorova et al.,



2019). Therefore, it was inferred that autochthonous POC displays no significant
relationship with $CH_4$ in this study.

The $CH_4$ concentrations in the two reservoirs were positively influenced by

terrestrial POC (Figure 8, 9). The structural equation model can well explain the
relationship with autochthonous POC ($R^2$=0.33 for mean value, $R^2$=0.35 for 50%
quantile) and $CH_4$ ($R^2$=0.53 for mean value, $R^2$=0.67 for 50% quantile), except for
terrestrial POC ($R^2$=0.13 for mean value, $R^2$=0.41 for 50% quantile). However,
terrestrial POC and $CH_4$ concentrations display a positive correlation in the spatial
dimension (Figure S10). The POC concentrations exhibit an exponential relationship
with the contribution of terrestrial POC, which may imply that terrestrial POC was a
major input for POC in the XLD and XJB reservoirs (Figure 10). Although the
autochthonous POC and $CH_4/CO_2$ mole ratios in March show a significantly positive
relationship mainly due to the outbreak of algal blooms, the most positive relationship
between terrestrial POC and $CH_4/CO_2$ mole ratios also indicates that the input of
terrestrial POC in the flood season is a crucial reason for $CH_4$ production (Figure S11).

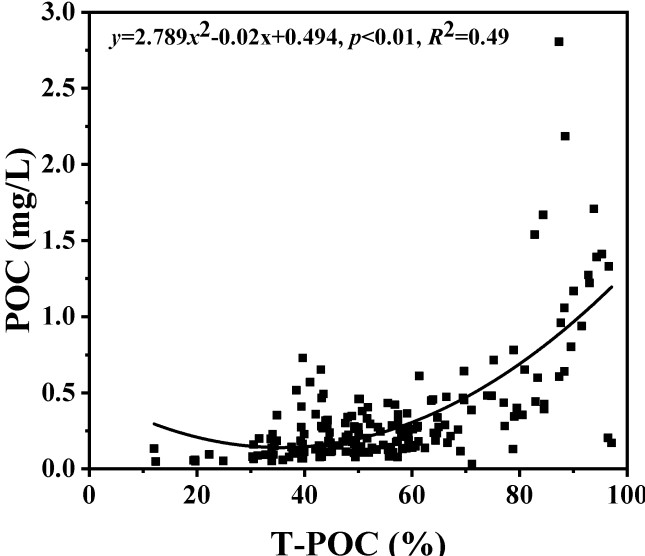

$$y=2.789x^2-0.02x+0.494, p<0.01, R^2=0.49$$


**Figure 10. Polynomial fit relationship between the contribution of terrestrial POC and POC**
**concentrations in the XLD and XJB reservoirs.**

The mass balance of $CH_4$ in the XJB reservoir was related to the estimation of $CH_4$



production. The annual $CH_4$ production showed an obvious single peak, and the highest
peak value was found in July (Figure 7B). However, there was low accumulation and no
significant increasing trend of terrestrial POC during the flood season influenced by
cascade damming, which indicates that no large amount of fresh terrestrial POC was
decomposed by bacteria to produce more $CH_4$. Water temperature significantly affects
the respiration rate of bacteria and then impacts the degradation of OM. Therefore, $CH_4$
production in the flood season may be affected by higher water temperatures, which
was supported by the structural equation model discussed above (Figure S8) and
consistent with previous studies (Delsontro et al., 2010; Beaulieu et al., 2014). However,
the difference in temperature from the XLD to XJB reservoirs in July and September
was not evident (Figure S3), which means that the $CH_4$ accumulation in deep
river-valley dammed cascade reservoirs is limitedly influenced by water temperature.
These results suggest that terrestrial POC was the key factor in $CH_4$ accumulation in the
XLD and XJB reservoirs.
However, there are numerous factors influence carbon migration and
transformation in damming rivers. More work needs to be done in the future, such as the
contribution of resuspended sediment POC to terrestrial POC. As complicated processes
of $CH_4$ accumulation are influenced by the input of terrestrial POC, more modeling
factors need to be considered.

## 561    5    Conclusions

POC in the XLD and XJB reservoirs mainly originates from terrestrial POC (56%).
$CH_4$ oxidation potentially influences the variations in $CH_4$ concentrations and $\delta^{13}C$-$CH_4$
in cascade reservoirs. Water level variations and flow regulation caused by reservoir
operation are primary factors for the input of terrestrial POC. The cumulative effect of
POC at a bimonthly scale was not significant in the XLD and XJB reservoirs. Terrestrial
POC displays more persistent impacts on $CH_4$ accumulation. This study provides a
scientific basis for revealing the major reason and mechanisms of $CH_4$ accumulation in
high-energy-density reservoirs under reservoir scheduling operation and helps to further



understand biogeochemical cycles of carbon in river-reservoir systems.

## Author contributions

YZ and YS contributed equally to this work, processed the data, drew the figures, designed the research framework and wrote the manuscript. ZL designed the study protocol, supervised the study, wrote and revised the manuscript. SG, LL, BZ and YQ carried out the field sampling and analyzed the samples.

## Competing interests

The authors declare that they have no conflict of interest.

## Acknowledgements

The National Natural Science Foundation of China (Project No. 51861125204, 51679226 and 42107273) primarily supported this work. The Interdisciplinary Team Project under auspices of "Light of West" Program from Chinese Academy of Sciences partially supported this work. Dr. Zhe Li is also supported by Chongqing Natural Science Funds for Distinguished Young Scientists (Project No. cstc2020jcyj-jqX0010). We thank the China Three Gorges Corporation for providing partial funding support for monthly sampling and daily hydrological data. We also thank Ms. Yinmin Xuan and Mr. Hailong Du who participated field sampling campaigns.

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
