# Peer review of "Methane accumulation affected by particulate organic carbon in upper Yangtze deep valley dammed cascade reservoirs, China"

_Biogeosciences, 2021_

## Author Comment (AC2)

**Response to Reviews for the manuscript #BG-2021-234**

We thank the Referee #2 and editor very much for holding and evaluating our manuscript entitled "Methane accumulation affected by particulate organic carbon in upper Yangtze deep valley dammed cascade reservoirs, China" (ID: BG-2021-234) and providing us with valuable feedbacks and comments. These comments are all useful and very helpful for revising and improving our manuscript. We prepared a new version of our manuscript according to Referee #2's comments and listed here point by point our replies. In the following, Referee #2's comments appear below in blue color, our detailed answers are in black and related modifications in the manuscript are reported in italics.

The main corrections in the paper and the responds to comments of Referee #2 are as following:

**Referee #2** addressed major and specific comments. Major comments"This is an interesting paper where the authors study the functioning of the hydropower reservoir on the accumulation and emission of methane. It's a timely contribution but some edition is needed before it should be accepted for publication. I am not English native and had a hard time understanding the writing particularly in the introduction. I think the writing in this ms should be carefully revised. I made some specific comments below but might be more. Most of the result section is discussion of the results. Finally, I do not clearly see from the results how the dam operation affects POC accumulation".

We agree with a hard time understanding the writing in the introduction which was modified in the revised manuscript. Moreover, we moved discussion of the results in results to discussion. And the dam operation affecting POC accumulation was explained at Lines 441-456 in the manuscript and Lines 394-408 in the revised manuscript.

Specific comments included 26 points.

1) Tittle: I think dammed is unnecessary to say since reservoirs are dammed....

We have no objection with what mentioned Referee #2. The tittle in this study was revised as "*Methane accumulation affected by particulate organic carbon in upper Yangtze deep valley cascade reservoirs, China*". Moreover, we deleted all "dammed" in the revised

manuscript.

2) Lines 56-58. Unclear why the cumulative effect of POC caused by cascade reservoirs should be at a bimonthly scale.

Because the sampling campaign was bimonthly conducted. The time scale of cumulative effect of POC is bimonthly scale.

3) Line 70. Revise the phrase "explicitly and well explained".

We understand the confusion which can be made and deleted "explicitly and" in the revised manuscript.

4) Lines 76-77. Unclear what you mean with "geomorphological alternation" and "and nutrients from surrounding communities".

We understand the confusion which can be made and deleted "geomorphological alternation", "*land use, large inputs of organic carbon (OC) and nutrients from surrounding cities*" was added at Lines 74-75 in the revised manuscript.

5) Line 79. Dam, reservoir, impoundment are synonymous to me⋯aren't they? Unclear what you mean with "Dam construction and reservoir impoundments".

In fact, dam construction and reservoir impoundments are not the same. For example, river-run-off hydropower projects construct dams along the river channel, but without creating reservoirs.

6) Line 90. You use supported very close twice. Unclear if you are commenting results from others studies or where does this evidence comes from since a reference is missing here.

Thanks for the comment. "Site-specific hydro-morphological characteristics, reservoir thermal regimes, and external OM loads from the upstream watershed supported consistent reservoir $CH_4$ emissions" was deleted and a reference was added at Line 86 in the revised manuscript.

7) Lines 93-94. Please revise "has been received challenges".

"has been received challenges" was revised as "*has been received attention*"at Line 89 in the revised manuscript.

8) Line 96. Predictors for what?

We understand the confusion which can be made and added "*for CH$_4$ emissions*" at Lines 91-92 in the revised manuscript.

9) Line 115. Unclear what you mean with "sophisticated".

We have no objection with what mentioned Referee #2 and deleted "appears to be sophisticated and site-specific and" at Line 110 in the revised manuscript.

10) Line 119 Correct "physiochemical".

"physiochemica" was corrected as "physio-chemical"

11) Line 121. Unclear what you mean with "dammed reservoirs".

We understand the confusion which can be made and deleted "dammed" at Line 116 in the revised manuscript.

12) Line 131. Which parameters?

We revised "parameters" as "*information*" at Line 126 in the revised manuscript.

13) Line 133. Unclear what you mean with every other month⋯monthly, bimonthly?

We understand the confusion which can be made, "every other month" means "bimonthly", so we added "bimonthly" at Line 128 in the revised manuscript.

14) Lines 138-139. Unclear what you mean here "The sampling time of a day was controlled between 8:00 AM and 6:00 PM"

We understand the confusion which can be made and revised as "*the sampling time of each sampling event was between 8:00 AM and 6:00 PM and limited between the dates of the 10th and 15th of that month*" at Lines 132-134 in the revised manuscript.

15) Line 142. Revise "disturbance to collected samples".

"tributary disturbance to collected samples" was revised as "*disturbance of tributary to collection of sample*s" at Line 137 in the revised manuscript.

16) Line 145-146. Two 10 L water samples? One 0.5 m below the water surface and other above the sediment? Which was the depth of the sampling points?.

We understand the confusion which can be made and added "*, respectively, with maximum water depth about 200 m and 120 m in XLD and XJB reservoirs*" at Lines 140-141 in the revised manuscript.

17) Line 148. How do you know the water column was well mixed? Did you measure water temperature?

The water column was well mixed because of high flow velocity and low water depth below dams.Therefore, we just measured temperature in the surface water and added "*because of high flow velocity and low water depth*" at Line 143 in the revised manuscript.

18) Line 152. Please define here HRT.

We defined "HRT" as "*the time required for the river water to be completely replaced per kilometer*" at Lines 148-149 in the revised manuscript.

19) Line 191. Particulate matter or chlorophyll a (Chl-a) concentration? I am not familiar with this methodology. Could you add any reference?

We understand the confusion which can be made, revised as "*For chlorophyll a (Chl-a), 500 mL water was filtered through Whatman® GF/C glass fiber membranes (Whatman®, UK), and then the fiber membranes were extracted by 90% cool acetone and measured spectrophotometrically for Chl-a concentration*" and added a reference at Lines 187-189 in the revised manuscript.

20) Line 195. Unclear what you mean with infiltration.

We understand the confusion which can be made and "the volume of water samples used infiltration" was revised as "*the volume of filtered water samples*" at Line 193 in the revised manuscript.

21) Line 220. Is it ok to use plankton as autochthonous end member? I think periphyton is the widely used community. At least write a couple of sentences regarding your decision.

We have no objection with what mentioned Referee #2, autochthonous organic matter derived from aquatic production, so all "plankton" in this study was corrected as "*periphyton*" and added a reference at Line 217 in the revised manuscript.

22) Line 302-303. This is discussion. Besides, why excrement? Do you mean raw sewage?

We deleted this sentence because of it is insignificant in this study.

23) Lines 312-325. Discussion.

Lines 304-325 of the manuscript was moved to Lines 349-371 of the revised manuscript.

24) Lines 335-337. Discussion.

Lines 335-337 of the manuscript was moved to Lines 551-552 of the revised manuscript.

25) Lines 374-388. Discussion.

Lines 374-388 of the manuscript was moved to Lines 514-527 of the revised manuscript.

26) Lines 389-411. Most of this section is discussion. Thus, move to discussion.

Lines 389-411 of the manuscript was moved to Lines 528-553 of the revised manuscript.

We also made some changes in the manuscript and Supporting Information listed in the following. These changes will not influence the content and framework of the paper.

1.  "*and low $CH_4$ concentration in deep river-valley cascade reservoirs are likely affected by $CH_4$ oxidation*" was revised at Lines 57-58 of the revised manuscript.

2.  "upstream organic carbon (OC) transport and sedimentation" was revised as "*OC sedimentation in upstream*" at Line 93 of the revised manuscript.

3.  "However" was deleted at Line 89 of the revised manuscript.

4.  "However" was changed as "*Moreove*r" at Line 101 of the revised manuscript.

5.  "elemental composition analysis of C, and N." was deleted and "*analysis of POC and particulate organic nitrogen (PON) by Elemental Analyzer (EA3000, Euro Vector®)*" was added at Lines 194-195 of the revised manuscript.

6.  "chlorophyll a" was deleted at Line 253 of the revised manuscript.

7.  "Figure 5" was revised as "*Figure 4*" at Line 336 of the revised manuscript.

8.  "Figure 4" was revised as "*Figure 5*" at Line 373 of the revised manuscript.

9.  "*The upper and lower ends of boxes represent the 0.25 and 0.75 percentiles, respectively. The upper and lower whisker caps represent the maximum and minimum values, respectively.*" was added and "middle black lines" was revised as "*solid black line in the box*" at Lines 411-413 of the revised manuscript.

10. "(the blue and red area represented confidence bands and confidence level for curves)" was revised as "*The blue and red area represent 95% confidence*

*intervals.*" at Lines 462-463 of the revised manuscript and in the Figure S8 of the revised Supporting Information.

11. "Figure S6" in the Supporting Information was moved and revised as "*Figure 11*" at Lines 554-556 of the revised manuscript.

12. "Figure S7-S11" was revised as "*Figure S6-S10*" in the Supporting Information.

13. "*The upper and lower ends of boxes represent the 0.25 and 0.75 percentiles, respectively. The upper and lower whisker caps represent the maximum and minimum values between 5% to 95%, respectively.*" was added and "middle black lines" was revised as "*solid white line in the box*" in the Figure S1 of the revised Supporting Information.

14. "*The upper and lower ends of boxes represent the 0.25 and 0.75 percentiles, respectively. The upper and lower whisker caps represent the maximum and minimum values, respectively.*" was added in the Figures S7 and S10 of the revised Supporting Information.